# Cloning and Expression Analysis of *HAT1* and *HDAC1* in the Testes of Mature Yaks and Their Sterile Hybrids

**DOI:** 10.3390/ani12162018

**Published:** 2022-08-10

**Authors:** Shijie Sun, Zhenhua Shen, Suyu Jin, Lin Huang, Yucai Zheng

**Affiliations:** College of Animal Husbandry and Veterinary Medicine, Southwest Minzu University, Chengdu 610041, China

**Keywords:** epigenetics, cattle-yak, male sterility, *HAT1*, *HDAC1*, H3K9 acetylation

## Abstract

**Simple Summary:**

Cattle-yak is the hybrid between male cattle (*Bos taurus*) and female yak (*Bos grunniens*). Male cattle-yak can not produce normal sperm. The mechanisms that underlie cattle-yak male sterility have not been elucidated. Histone acetylation is a common regulation mode that plays an important role in the development of gametes. The objective of this study was to explore the molecular mechanism of male sterility in yak hybrids based on histone acetyltransferase 1 (*HAT1*) and histone deacetylase 1 (*HDAC1*), two enzymes that regulate histone acetylation. The mRNA and protein expression levels of *HAT1* in the testes of adult cattle-yaks were significantly lower than in adult yaks, and the protein expression levels of *HDAC1* were significantly higher than in yaks. In addition, H3K9 acetylation levels in cattle-yak testes were significantly lower than in yaks. These results suggest that male sterility in cattle-yaks might be associated with decreased histone acetylation levels in the testes.

**Abstract:**

The objective of this study was to explore the molecular mechanism of male sterility in yak hybrids based on *HAT1* and *HDAC1*. Total RNA was extracted from the testes of adult yaks (*n* = 11) and sterile cattle-yaks (*n* = 11) followed by reverse transcription. The coding sequence (CDS) of yak *HAT1* and *HDAC1* were obtained by conventional polymerase chain reaction (PCR) and gene cloning. The testicular mRNA and protein levels of *HAT1* and *HDAC1* in yaks and cattle-yaks were detected by quantitative PCR (qPCR) and Western blotting, respectively, and the histone H3 lysine 9 (H3K9) histone acetylation level in the testes of yaks and cattle-yaks was assayed using enzyme linked immunosorbent assay (ELISA). The results showed that the CDS of *HAT1* and *HDAC1* were 1242 bp and 1449 bp in length, encoding 413 and 482 amino acids, respectively; yaks had a similar mRNA sequence as cattle in both genes. The testicular mRNA and protein levels of *HAT1* of cattle-yaks were significantly lower than those of yaks, and the protein level of *HDAC1* was significantly higher than that of yaks. ELISA showed that the acetylation level of testicular H3K9 was significantly lower in yak hybrids than that of yaks. The present results suggest that the decreased level of *HAT1* and increased level of *HDAC1* may result in the decreased H3K9 acetylation in cattle-yaks and might be associated with their sterility.

## 1. Introduction

Yak (*Bos grunniens*) is a unique cattle species adapted to the harsh conditions of the alpine grassland, such as hypoxia and insufficient forage. As a unique genetic resource, yaks can make full use of the grassland of the Qinghai-Tibet Plateau, providing products such as meat and milk for the local farmers [1]. The cattle-yak, an interspecific hybrid of yaks and cattle, has obvious heterosis, showing higher production performance. However, the male hybrids are sterile and thus affect the breeding of yaks [2]. It has been reported that the male sterility of cattle-yaks is mainly due to abnormal spermatogenesis [3,4].

Many genes related to spermatogenesis have been studied in the testes of yaks and cattle-yaks, such as *Cdc2, Mei1, Prdm9, SYCP2, SYCP3*, and other genes possibly involved in the regulation of sperm production [2,5,6,7,8], most of which showed significantly decreased mRNA levels in cattle-yak testes [9,10]. During mitosis and meiosis of mammalian spermatogenesis, the DNA and histones of spermatogenic cells undergo epigenetic modification, which affects the expression and activation of specific genes and makes spermatogenesis under strict regulation [4]. Epigenetic regulation refers to the heritable changes in gene expression under the condition that the DNA sequence in the original genome does not change. It only affects in gene expression and not the DNA sequence of the gene [11]. Acetylation modification of histones is an important mechanism of gene epigenetic transcription regulation, which is mainly catalyzed by histone acetylases (HATs) and histone deacetylases (HDACs) [12,13,14]. HATs add hydrophobic acetyl groups to the amino-terminal amino acid residues of histones to deduce the interaction between DNA and histones, loosen the chromatin structure, and facilitate gene transcription, while HDACs deacetylate histones and, thus, inhibit transcription [14,15].

Histone H3 acetylation plays a crucial role in certain stages of spermatogenesis [16]. It has been demonstrated that acetylated H3K9 and H3K18 of mouse histones have dynamic changes during the division and differentiation of germ cells [17,18]. Thus, we hypothesized that testicular *HAT1* and *HDAC1* may be related to the hybrid male sterility of yaks. The purpose of this study was to explore the possible relationship between the regulation of histone acetylation and male sterility of cattle-yaks. In the present study, the *HAT1* and *HDAC1* genes of yak were cloned, and their mRNA and protein levels, as well as H3K9 acetylation in the testes, were compared between sterile cattle-yaks and normal yaks.

## 2. Materials and Methods

### 2.1. Ethical Approval

All animal procedures were performed according to protocols approved by the Institutional Animal Care and Use Committee of Southwest Minzu University (No. 2019-0032).

### 2.2. Animals and Sampling

The adult male Maiwa yaks (*Bos grunniens*, n = 11) and the male cattle-yaks (F1 hybrids between male cattle and female Maiwa yaks, n = 11) in this experiment were provided by a commercial slaughterhouse in Qingbaijiang region, Sichuan Province, aged at about 4 to 6 years old. All the yaks and cattle-yaks were raised in the same alpine pasture before slaughter. The testes and epididymides were collected immediately after the experimental animals were slaughtered. The testes were cut into several parts and promptly frozen in liquid nitrogen and transferred to laboratory, stored at −80 °C before analysis. Epididymis was fixed with tissue fixative and brought back to the laboratory. The experiment was conducted in accordance with the Regulation on the Administration of Laboratory Animals (2017, China State Council).

### 2.3. Sectioning and HE Staining of Yak and Cattle-Yak Epididymis

Histological sections of yak and cattle-yak epididymis were prepared by conventional methods and stained with hematoxylin-eosin staining (HE). The sections were observed with an Olympus light microscope, mainly to observe whether there were sperm in the epididymis.

### 2.4. RNA Extraction and Reverse Transcription

Total RNA was extracted from the testes of yaks and cattle-yaks with Trizol reagent (Invitrogen, Carlsbad, CA, USA). The Revert Aid First Strand cDNA Synthesis Kit (Thermo, Waltham, MA, USA) was used in reverse transcription according to the manufacturer’s instructions. The reaction system of first strand cDNA synthesis included total RNA, Oligo (dT)18 primer, Random Hexamer primer, RNase-free water, 5× Reaction buffer, RiboLock RNase inhibitor, 10 mM dNTP Mix, and RevertAid M-MuLV RT. The reverse transcript products were stored at −20 °C.

### 2.5. Cloning and Sequencing of the HAT1 and HDAC1 Genes of Yaks

All primers and conditions of PCR in this experiment are shown in Table 1. The clone program of *HAT1* and *HDAC1* was similar. Super Taq DNA polymerase (GeneCopoeia, Rockville, MA, USA) was used for all PCR reactions. The program of PCR was 94 °C for 5 min; 35 cycles of 94 °C for 30 sec, 59 °C for 30 sec, and 72 °C for 4 min; 72 °C for 7 min. The PCR products were separated with 1.5% agarose gel electrophoresis and purified by a DNA purification kit (Omega, Norcross, GA, USA), followed by cloning into pMD19-T vector (TaKaRa, Shiga, Japan).

### 2.6. Analysis of HAT1 and HDAC1 mRNA Expressions by Quantitative PCR

Quantitative PCR was used to compare the mRNA levels of *HAT1* and *HDAC1* in the testes of adult yaks and cattle-yaks. Primer sequences are listed in Table 1. The 18s rRNA was used as the internal reference gene [19]. The 25 µL reaction contained 1 µL of cDNA, 12.5 µL of TB Green PreMix Ex Taq II (TaKaRa, Shiga, Japan), 1 µL each of forward and reverse primers (from 10 µmol/L of stock), and ultra-pure water. PCR was performed using the CFX96 real-time PCR detection system (Bio-Rad, Hercules, CA, USA). The PCR conditions for *HAT1* and *HDAC1* were set as the following: one cycle of 30 s at 95 °C followed by 40 cycles of 5 s at 95 °C and 1 min at 60 °C. The PCR conditions for 18S rRNA were set as the following: one cycle of 1 min at 95 °C followed by 40 cycles of 15 s at 95 °C, 20 s at 59 °C, and 30 s at 72 °C. The melting curve was analyzed from 65 °C to 95 °C with each increase of 0.5 °C in the plate reading. Each sample in the analysis was technically duplicated.

### 2.7. Analysis of HAT1 and HDAC1 Protein Level by Western Blotting

The testes of yaks or cattle-yaks were lysed in RIPA buffer (150 mM NaCl, 50 mM Tris-HCl, 1% NP-40, 0.5% deoxycholate, 0.1% SDS, pH8.0) (Solarbio, Beijing, China) containing 1 mM of phenylmethanesulfonyl fluoride (PMSF) (Solarbio, Beijing, China). The resulting samples were separated by 12% SDS-PAGE and transferred to PVDF membranes using a semi-dry system at 40 mA. The membranes were blocked in 5% skim milk at 25 °C for 1 h, followed by incubation with the specific primary antibody overnight at 4 °C. The primary antibodies were *HAT1* (Cambridge, UK, abcam, 1:3000), *HDAC1* (Cambridge, UK, Abcam, 1:1000), and GAPDH (Affinity, Jiangsu, China, 1:3000). Then the membranes were incubated with horseradish peroxidase-conjugated goat-anti-rabbit IgG (Affinity, Jiangsu, China, 1:10,000) for 1 h at 37 °C. The signal intensities were measured using Clarity Western ECL Substrate (Bio-Rad, Hercules, CA, USA) and image analysis software (ImageJ1.8.0, NIH, Bethesda, MD, USA).

### 2.8. Quantification of H3K9 Acetylation Level by ELISA

Total histones were extracted from frozen testes of yaks and cattle-yaks with the EpiQuik™ Whole Histone Extraction Kit (Epicentek, Farmingdale, NY, USA) following the manufacturer’s protocol. The EpiQuik Global Histone H3 Acetylation Assay Kit (Epicentek, Farmingdale, NY, USA) was used to quantify H3K9 according to the manufacturer’s instructions.

### 2.9. Statistical Analysis

All data were analyzed using SPSS 22.0 (IBM, Armonk, NY, USA) and results are expressed as mean ± standard error of mean (SEM). GraphPad Prism 8.01 (GraphPad Software, San Diego, CA, USA) was used to integrate data and make graphs for visual representation of the data. The cycle threshold (Ct) resulting from RT-PCR was analyzed using the 2-∆∆Ct method [19]. The data were tested for normality using a Kolmogorov–Smirnov test (*p* ≥ 0.05). Two-tailed Student’s t-test was used to test for differences. Significant differences were considered to exist when the *p*-value was less than 0.05.

## 3. Results

### 3.1. Histological Comparison of Yak and Cattle-Yak Epididymis

In the epididymal sections of yak, mature spermatozoa were observed in the epididymal duct (Figure 1A). However, no sperm were observed in the epididymal duct of cattle-yaks (Figure 1B).

### 3.2. Cloning and Sequencing of Yak HAT1 and HDAC1 Genes

The CDS sequences of yak *HAT1* and *HDAC1* genes were cloned from yak testis.

The CDS of the yak *HAT1* gene was 1242 bp as revealed by sequencing (GenBank accession No. MW300349), coding 413 amino acids. This sequence was basically the same as those of predicted *HAT1* variant 2 of wild yak (XM_005887571.1 and XM_005887572.2, variant 2 has ATGGCGGGATTTGGTGCT at position 1 to 18) and cattle *HAT1* gene (NM_001034347.1) in the GenBank, except one nucleotide difference at position 444 of nucleotide sequence (T to A), which resulted in no amino acid mutation. The CDS of yak *HAT1* gene shares 96.54% to 99.92% sequence similarity with six mammal species, namely, wild yak and cattle (both 99.92%), sheep (99.28%), camel (97.10%), pig (96.86%), and horse (96.54%).

The CDS of the *HDAC1* gene was 1449 bp in length (GenBank accession No. MW300348), coding 482 amino acids. The *HDAC1* gene sequences of yak and predicted wild yak (XM_005906205.1) are similar, with only one nucleotide difference (yak: G, cattle: A) at position 1110 with cattle (NM_001037444.2), resulting in no amino acid mutation. The CDS of *HDAC1* gene shares 95.31% to 100.00% sequence similarity with six mammal species, namely, wild yak (100.00%), cattle (99.93%), sheep (98.69%), pig (96.48%), horse (95.93%), and camel (95.31%).

### 3.3. HAT1 and HDAC1 Expression in the Testes of Yaks and Cattle-Yaks

The mRNA levels of *HAT1* and *HDAC1* in the testes of yaks and cattle-yaks were assayed by qPCR with specific primers, and 18S rRNA was used as the reference gene. The results showed that the relative mRNA levels of the *HAT1* gene in the testes of cattle-yaks were significantly lower (*p* < 0.05) than those in yaks; however, the testicular mRNA levels of the *HDAC1* gene were not significantly different between yaks and cattle-yaks (Figure 2A,B).

The expressions of *HAT1* and *HDAC1* proteins in testes were tested by Western blot (Figure 2C,D), and GAPDH was used as the reference. The results showed the expression of *HAT1* in yaks was significantly higher (*p* < 0.01) than in cattle-yaks, while for *HDAC1*, the expression in yaks was significantly lower (*p* < 0.05) compared to cattle-yaks.

### 3.4. The Acetylation Level of H3K9 in the Testes of Yaks and Cattle-Yaks

The ELISA results showed that the acetylation level of H3K9 was decreased in the testes of cattle-yaks compared with yaks (*p* < 0.01) (Figure 3).

The cattle-yaks had significantly decreased *HAT1* at both mRNA and protein levels, and significantly higher *HDAC1* at protein level. The cattle-yaks had significantly lower testicular H3K9 acetylation level than yaks.

## 4. Discussion

Until now, the molecular mechanism of hybrid male sterility of cattle-yaks has not been elucidated. In previous studies, it has been found that the numbers of germ cells in cattle-yak testes and epididymides were significantly less than that in yak and there were almost no mature sperm, suggesting that the blocked period of bovine spermatogenesis was likely to be at the primary spermatocyte stage [2,20,21]. Sperm was found in the epididymal duct of yak. However, in the cattle-yak counterpart, there were no sperm in the epididymal canal. It has been reported that bovine spermatogenesis is blocked at the primary spermatocyte stage [2,22] In our previous study, we found that the expression level of PRDM9 in cattle yaks was different from that in yaks, the expression of PRDM9 was lower in cattle-yak testes [2]. We then detected and analyzed the methylation level of yak and cattle-yak testes, and the change was also consistent with the change trend of PRDM9 [20]. We speculated that acetylation was also involved in the regulation of bovine spermatogenesis disorder and demonstrated this in the current study.

Histone acetyltransferase can regulate histone acetylation, change chromatin configuration, and activate or enhance gene expression by adding acetyl groups to histones. *HAT1* is an evolutionarily conserved type B histone acetyltransferase [23]. The regulatory sites confirmed by research are H4K5 and H4K12, and some studies have also shown that the acetylation level and of histone H3 are affected by *HAT1* [24]. Previous studies have shown that *HAT1* also plays an important role in cell proliferation, histone production, and glucose metabolism [25]. The decrease of *HAT1* expression level could lead to the delay of G2/M phase and abnormal apoptosis [23], which would affect the transformation of germ cells to spermatogonia and reduce the number of spermatogonia and primary spermatocytes. The decrease of *HAT1* mRNA and protein levels in cattle-yak testes probably led to the insufficient number of bovine spermatogonial stem cells and primary spermatocytes. The change of *HAT1* expression level in the testes could lead to the arrest of cattle-yak spermatogenesis by reducing the expression of genes, such as *Cdc2* [26], that play an important role in differentiation regulation during spermatogenesis. *HAT1* may be related to cattle-yak male sterility, which is worthy of further study.

*HDAC1* is a catalytic subunit of a variety of protein complexes, which can inhibit gene transcription and participate in the regulation of cell proliferation, cell cycle, and cell differentiation [27]. There was no significant difference between yak and bovine *HDAC1* at the mRNA level, but the trend was consistent with that at the protein level. Some studies have shown that *HDAC1* plays an irreplaceable role in the early stage of embryonic development, and the deletion of *HDAC1* will lead to embryonic death [28]. Other studies have also shown that *HDAC1* is involved in the regulation of mammalian epithelial and stem cell differentiation [29]. The increased expression of *HDAC1* in cattle-yak testes might limit the division and differentiation of primordial germ cells limiting numbers of spermatogonia and could prevent the production of enough spermatogonia. The relationship between *HDAC1* and cattle-yak sterility deserves further study.

H3K9 acetylation is expressed in many stages of mammalian spermatogenesis, but the expression varies at different stages [30]. In the spermatogonia stage, the acetylation level of H3K9 is high. Spermatogonia are a type of cell with strong division ability, and spermatocytes are produced by the division and differentiation of spermatogonia [17]. In this study, the acetylation level of H3K9 in cattle-yak testes was decreased compared to yak testes, which might lead to the inhibition of genes related to reproductive development regulated by H3K9 [24]. *Cdc2* is also a key gene responsible for G2/M regulation and is regulated by H3K9 acetylation [26]. The decrease of the H3K9 acetylation level would lead to the decrease of *Cdc2* transcriptional activity, inhibit the mitosis of spermatogonia, and fail to produce a sufficient number of spermatogonia, which could directly lead to the decrease of the number of spermatocytes.

## 5. Conclusions

This study showed that in the testes of sterile cattle-yak, the mRNA and protein levels of *HAT1* were significantly lower than those of yaks, and the protein level *HDAC1* was higher than that of yak. The level of H3K9 acetylation in cattle-yak testes was lower than that of yak. The decrease in the level of acetylated testes in cattle-yak might be related to their sterility. These results suggested the molecular mechanisms of cattle-yak male sterility from an epigenetic perspective. However, there are many types of cells in testes, including germ cells at different stages of development, epithelial cells, support cells, and mesenchymal cells. Therefore, further detailed analysis for different cells is needed.

## Figures and Tables

**Figure 1 animals-12-02018-f001:**
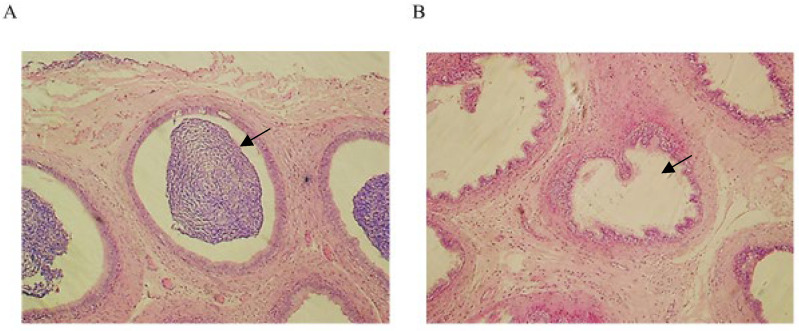
Histological examination of the epididymis in yaks and cattle-yaks. Magnification: 40× (images (**A**,**B**)). (**A**) Sections of HE-stained yak epididymis. (**B**) Sections of HE-stained cattle-yak epididymis. The arrow points to the epididymal canal.

**Figure 2 animals-12-02018-f002:**
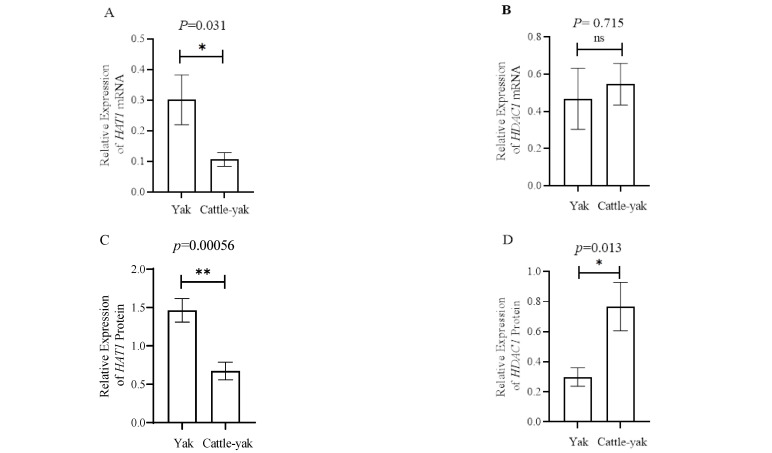
*HAT1* and *HDAC1* expression in the testes of yaks and cattle-yaks. (**A**) The relative expression of the *HAT1* mRNA detected by real-time PCR; 18s rRNA was used as reference genes. (**B**) The relative expression of the *HDAC1* mRNA detected by real-time PCR; 18s rRNA was used as reference genes. (**C**) The relative expression of the *HAT1* protein between yaks and cattle-yaks. (**D**) The relative expression of the *HDAC1* protein between yaks and cattle-yaks. (**E**) The expression of *HAT1* and *HDAC1* proteins detected by Western blotting. GAPDH was used as reference gene. Lanes 1–4 were yaks and lanes 5–8 were cattle-yaks. * *p* < 0.05 and **: *p* < 0.01.

**Figure 3 animals-12-02018-f003:**
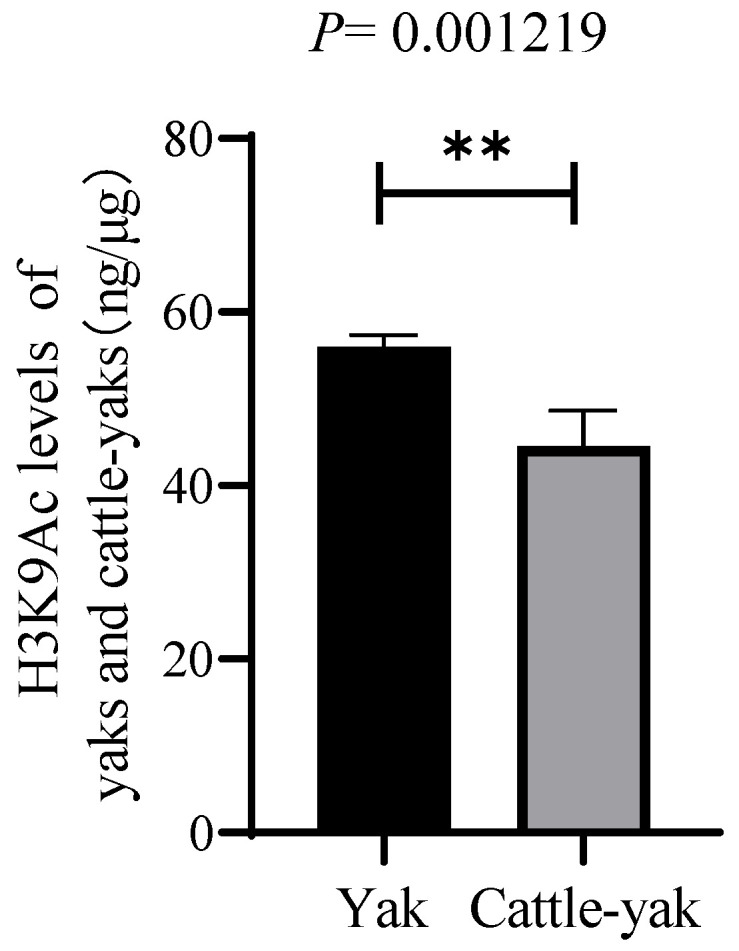
The levels of H3K9 acetylation in the testes of yaks and cattle-yaks; **: *p* < 0.01.

**Table 1 animals-12-02018-t001:** Primer information in this experiment.

GenBank Accession No.	Gene Name	Primer Sequence (5´–3´)	Annealing Temperature (°C)	Product Size (bp)
NM_001034347.1	*HAT1*	F: TCGGAAATGGCGGGTTTGA R: AAGGGAAGTAATTGCAGTGGTA	60	1559
NM_001037444.2	*HDAC1*	F: GGACCGATTGACGGGAGGG R: GGGTTCAAGAGTTTGGGAGGG	60	1699
NM_001034347.1	*HAT1*	F: AGCCTATCAACAATGTTCCGTG R: AGCTTCTTTTTCCAGCAACG	60	159
NM_001037444.2	*HDAC1*	F: CCAGTGCAGTTGTCTTGCAG R: ACGAATGGTGTAGCCACCT	60	158
NR 036642	18S rRNA	F: CTGAGAAACGGCTACCACATC R: CAGACTTGCCCTCCAATGG	59	168

## Data Availability

The data are available on request from the corresponding author.

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
