# Peer review of "Cloning and Expression Analysis of HAT1 and HDAC1 in the Testes of Mature Yaks and Their Sterile Hybrids"

_animals, 2022, doi:10.3390/ani12162018_

Round 1

Reviewer 1 Report

Aiming at elucidating the molecular mechanism of male sterility in cattle-yak hybrids, the authors noticed some expression differences in the genes HAT1 and HDAC1 between testes containing viable and apoptotic sperm. This observation has a little value as dying cells often give false readouts. Moreover, as detailed in Major Concerns, Introduction, Animals, Experimentation are improper.

Major Concerns,

  1. The major players in the hybrid sterility are not introduced according to their recorded relevance. PRDM9 is the only speciation gene found so far in mammals. Incompatibility of PRDM9 does fit the observed male sterility phenomenon. The authors should examine this before suggesting other mechanisms.
  2. The selected sample can not elucidate the molecular mechanism of male sterility in cattle-yak hybrids. Much better selection would be the F2 hybrid. F2 can be readily produced since F1 female hybrids are fertile. Using genetic mapping of male sterility in F2, it would be easy to point to the underlying gene and to examine the molecular mechanism of male sterility using a better background.
  3. Experimentation is improper as the author did not select against dying cells.

Minor Remarks:

  1. Gene name should be italicized all along.
  2. The authors should exercise care in introducing spaces between words and sentences all along: E.g. Line 168 “butrelative” should be “but relative”.

Author Response

Major Concerns,

1.The possible important functions of PRDM9 in the hybrid sterility of cattle-yak have been introduced, followed by other possible mechanisms.

2.This experiment was to explore the expression differences of the two important genes and to further discuss the mechanism of hybrid male sterility. In fact, mainly F1 hybrids of yaks are used due to their economic values, while F2 hybrids shows lower meat and milk performance. So testis samples of F2 hybrids are not easily obtained. The proposal is very important and helpful, and we plan to conduct the proposed methods.   

3.This experiment was designed to assay gene expression of the whole testis, so we did not separate any types of cells in the testis, thus this is a preliminary study.

Minor Remarks:

Both the first and second points have been modified.

Reviewer 2 Report

Dear Authors,

I reviewed your manuscript and I have some suggestions to improve the comprehension of the paper. 

There are some shortcomings in materials and methods that could affect the repeatability of the experiment:

Table 1 - I checked the primers of the second fragment of HAT1 (159 bp) and I supposed they are inverted. I think it could be possible that you had reversed the forward, and vice versa for the reverse. In addition, it should be specified why there are 2 fragments for each target gene. It could be added a column to specify qPCR or cloning, or give a “name” to the fragments and, then, you could report it in the corresponding part of the text. 

Line 104-106 – The temperature and time for extension lack in PCR conditions for HAT1 and HDAC1

Line 174-176. It is a part of the text of the template, delete it please

Figure 2 – I suggest to split in 2 figures, the first for the mRNA expression by qPCR, with the 4 graphs, (A, B, C, D), and the second for protein expression by Western blotting. Consequently, the text should be updated.

As shown in the part B of figure 2, and in lines 166-169 of the text, you affirm that “mRNA levels of the HDAC1 gene in the testes of yaks or cattle–yaks were not significantly different”. In the discussion section these findings should be discussed (lines 229-238)

Author Response

由于粘贴误差,HAT1定量PCR引物的f端和r端确实被回填。

荧光定量PCR是一个两步过程,再饱和延伸在同一步

图2指的是其他动物文章中的数字格式。目前尚不确定是否需要拆分。

Reviewer 3 Report

The manuscript Cloning and expression analysis of HAT1 and HDAC1 in the testes of mature yaks and their sterile hybrids is interesting, fairly well-written, and relevant to the readers of Animals.

Suggest refraining from referring to the hybrids as "sterile" since this was not established in the present study. Certainly the introduction should contain the references in the literature that suggests these hybrids are infertile.

Other considerations.

Line 10. Confusing and not "all" hybrids have been tested for sperm capacity. Suggest "Male cattle-yak hybrids lack sperm production capacity"

Line 29. Suggest "had similar RNA sequence as cattle"

Line 32. Suggest "lower [in yak hybrids] than yaks.

Line 158. Suggest "similar" rather than "same"

Line 198. Delete "extremely significant"

Line 235-237. Suggest "The [increased] expression of HDAC1 in testes of [yak-]cattle hybrid might....germ cells [limiting numbers of spermatogonia].

Line 240. Suggest "varies at different stages" rather than "levels are different in"

Line 243. Suggest "decreased" rather than "significantly lower"

Author Response

With regard to  "sterile" , we have added he staining sections of epididymal tissue, which could prove that the male F1 generation of cattle is sterile.

About other considerations,we think your suggestion is very correct and have made corresponding modifications

Round 2

Reviewer 1 Report

The authors declare in their response "1. The possible important functions of PRDM9 in the hybrid sterility of cattle-yak have been introduced, followed by other possible mechanisms." Yet, I do not see such modification in the revised manuscript. The authors themselves declare that their results and design are preliminary. As I have stated in my previous review it is well documented that PRDM9 is responsible to hybrid-cattle sterility. Its mode of action is also related to histone modifications. Thus, it is rather expected that downstream genes would be affected. To prove that HAT1 and HDAC1 genes are responsible the authors should explore their genomic sequence, which is available for both species. I guess that with such investigation they would be convinced that that the sterility problem arises of variation of the zinc fingers of PRDM9. They should read current literature such as https://doi.org/10.1016/j.tig.2021.06.008. Moreover, they do not cite relevant paper that also shows that histone modifications are developmentally dysregulated in cattle-yak (https://doi.org/10.1016/j.theriogenology.2020.01.001). They can follow this last example to realize how to proceed with a similar paper that does not ignore the known literature.

Author Response

Thank you for your correction.

We have done research on PRDM9 before, and the results have been published. At the end of the first paragraph of the discussion part, the previous results of PRDM9 and the reasons for acetylation are added.